# Assessment of the outbreak risk, mapping and infection behavior of COVID-19: Application of the autoregressive integrated-moving average (ARIMA) and polynomial models

Hamid Reza Pourghasemi[1]*, Soheila Pouyan[1], Zakariya Farajzadeh[2], Nitheshnirmal Sadhasivam[3], Bahram Heidari[4]*, Sedigheh Babaei[1], John P. Tiefenbacher[5]

1 Department of Natural Resources and Environmental Engineering, College of Agriculture, Shiraz University, Shiraz, Iran, 2 Department of Agricultural Economics, College of Agriculture, Shiraz University, Shiraz, Iran, 3 Department of Geography, School of Earth Science, Bharathidasan University, Tiruchirappalli, Tamil Nadu, India, 4 Department of Plant Production and Genetics, School of Agriculture, Shiraz University, Shiraz, Iran, 5 Department of Geography, Texas State University, San Marcos, Texas, United States of America

* hr.pourghasemi@shirazu.ac.ir (HRP); bheidari@shirazu.ac.ir (BH)

## Abstract

Infectious disease outbreaks pose a significant threat to human health worldwide. The outbreak of pandemic coronavirus disease 2019 (COVID-19) has caused a global health emergency. Thus, identification of regions with high risk for COVID-19 outbreak and analyzing the behaviour of the infection is a major priority of the governmental organizations and epidemiologists worldwide. The aims of the present study were to analyze the risk factors of coronavirus outbreak for identifying the areas having high risk of infection and to evaluate the behaviour of infection in Fars Province, Iran. A geographic information system (GIS)-based machine learning algorithm (MLA), support vector machine (SVM), was used for the assessment of the outbreak risk of COVID-19 in Fars Province, Iran whereas the daily observations of infected cases were tested in the—polynomial and the autoregressive integrated moving average (ARIMA) models to examine the patterns of virus infestation in the province and in Iran. The results of the disease outbreak in Iran were compared with the data for Iran and the world. Sixteen effective factors were selected for spatial modelling of outbreak risk. The validation outcome reveals that SVM achieved an AUC value of 0.786 (March 20), 0.799 (March 29), and 86.6 (April 10) that displays a good prediction of outbreak risk change detection. The results of the third-degree polynomial and ARIMA models in the province revealed an increasing trend with an evidence of turning, demonstrating extensive quarantines has been effective. The general trends of virus infestation in Iran and Fars Province were similar, although a more volatile growth of the infected cases is expected in the province. The results of this study might assist better programming COVID-19 disease prevention and control and gaining sorts of predictive capability would have wide-ranging benefits.

**Data Availability Statement:** All relevant data are within the manuscript and its Supporting Information files.

**Funding:** A grant with number 96GRD1M271143 was initially defined by Shiraz University. The funder had no role in study design, data collection and analysis, decision to publish, or preparation of the manuscript.

**Competing interests:** The authors have declared that no competing interests exist.

## Introduction

In December 2019, several pneumonia infected cases were reported in Wuhan, China [1, 2]. In January 2020, a novel coronavirus (2019-nCoV) that was later formally named COVID-19 was approved in Wuhan [3]. It was announced that the disease is a severe acute respiratory syndrome coronavirus 2 (SARS-CoV-2). The virus elevated concerns within China as well as the global community as it was believed to be transmitted from human to human [4]. Initially, China witnessed the largest outbreak in Hubei and other nearby provinces. The spread in China was controlled soon thereafter through stringent preventive measures, but other parts of the world (Europe, the Middle East, and the United States) were increasingly affected by the outbreak through transmission by infected travellers from China. A similar outbreak soon followed in other Asian countries [5]. Its global spread to more than 150 countries led to the declaration in mid-March 2020 that COVID-19 was a pandemic [6]. By June 18, 2020, there were nearly 8.60 million cases worldwide, with 455575 deaths attributed to COVID-19 [7]. Currently, the United States (2263651), Brazil (983359) and Russia (561091) have the largest number of confirmed cases, whilst the United States (120688), Brazil (47869) and UK (42288) have the highest number of casualties, respectively [7, 8]. Iran with 197647 recorded cases and 9272 deaths is the most affected country in the Middle East (as of June 18, 2020) and infected cases are expected to surge in the coming days [7, 9]. The outbreak of COVID-19 has disrupted and depressed the world economy, whereas Iran is among the most severely affected by massive economic losses, largely compounded by politically motivated sanctions imposed by other governments [10]. The problem has been exacerbated as no specific medicine is yet realized for COVID-19 disease treatment, though there are a few pre-existing drugs that are being tested, so regions are presently concentrating their efforts on maintaining the infection rate in a level that assists in reducing virus spread [11]. This has led to most states imposing lockdowns, encouraging social distancing, and restricting the sizes of gatherings to limit transmission [12]. There is a pressing necessity for scientific communities to aid governments in their efforts to control and prevent transmission of the virus [13].

During previous virus outbreaks stemming from Zika, influenza, West Nile, Dengue, Chikungunya, Ebola, Marburg, and Nipah, geographic information systems (GISs) have played significant roles in providing significant insight via risk mapping, spatial forecasting, monitoring spatial distributions of supplies, and providing spatial logistics for management [13]. In this current situation, risk mapping is critical and may be used to aid governments' need for tracking and management of the disease as it spread in places with the highest risk. Sánchez-Vizcaíno et al. [14] used a multi-criteria decision making (MCDM) model to map the risk of Rift Valley fever in Spain. Traditional statistical techniques had also been used to detect the risk of an outbreak [14]. Reeves et al. [15] employed an ecological niche modelling (ENM) technique for mapping the transmission risk of MERS-CoV; the Middle Eastern name for the coronavirus known as SARS-CoV-2. Similar techniques have been in the Nyakarahuka et al. [16] study to map Ebola and Marburg viruses risks in Uganda. They assessed the importance of environmental covariates using the maximum entropy model.

More recently, the use of machine learning algorithms (MLAs) for mapping the risk of transmission of viruses has been increasing which is due to the demonstrated superior (and more accurate) predictive abilities of the MLA models over traditional methods [17]. Jiang et al. [18] employed three MLAs–backward propagation neural network (BPNN), gradient boosting machine (GBM), and random forest (RF)–to map the risk of an outbreak of Zika virus. Tien Bui et al. (2019) compared different MLAs–artificial neural network (ANN) and support vector machine (SVM) with ensemble models including adaboost, bagging, and random subspace–for modelling malaria transmission risk. Similarly, GBM, RF, and general

additive modelling (GAM) were used by Carvajal et al. [19] to model the patterns of dengue transmission in the Philippines. Mohammadinia et al. [20] employed geographically weighted regression (GWR), generalized linear model (GLM), SVM, and ANN to develop a forecast map of leptospirosis; GWR and SVM produced highly accurate predictions. Saba and Elsheikh [21], also used the nonlinear autoregressive ANN model to forecast COVID-19 outbreak. Another statistical-based model that recently has been applied to forecast the behaviour of COVID-19 outbreak and death cases is ARIMA in which the forecast process is as a function of time. Recently, the significant ability of this model to forecast COVID-19 outbreak in Egypt [21] and coronavirus related deaths in Iran [22] has been reported. Benvenuto et al [23] performed ARIMA model on the Johns Hopkins epidemiological data and they found that the spread of virus tends to be slightly decreasing. However, Ahmar and del Val [24] combined the α-Sutte indicator with ARIMA and developed a model to forecast COVID-19 outbreak in Spain. Their combined model presented more accurate forecast compared to the ARIMA model.

The literature shows that very few studies have tried to use GIS for analysis of COVID-19 outbreak in human communities. Kamel Boulos and Geraghty [25] described the use of online and mobile GIS for mapping and tracking COVID-19 whilst Zhou et al. [13] revealed the challenges of using GIS for SARS-CoV-2 big data sources. To our knowledge, there has been no study with a focus on mapping the outbreak risk of the COVID-19 pandemic. The aims of the present study were to analyze the risk factors of coronavirus outbreak and test the SVM model for mapping areas with a high risk of human infection with the virus in Fars Province, Iran. In addition, the growth trend of the COVID-19 infestation in Fars Province was analyzed and compared with the growth rate (GR) of Iran and several other countries. The outcome of the present study lays a foundation for better planning and understanding the factors that accelerate the virus spread for use in disease control plans in human communities. The methodology of this research can be used for mapping the outbreak risk of COVID-19 and for detecting the trend of COVID-19 infections in other parts of the world. This study also can aid local authorities in imposing strict social distancing measures in the regions with high outbreak risk. Furthermore, this study can be helpful in determining the significant effective factors that influence the COVID-19 outbreak risk.

## Materials and methods

### Study area

The study area is in the southern part of Iran with an area of 122608 square kilometres located between $27°2'$ and $31°42'$ N and between $50°42'$ and $55°36'$ E. Fars is the fourth largest province in Iran (7.7% of total area) with a population density of 4851274 (based on in 2016 report). Fars Province is divided into 36 counties, 93 districts, and 112 cities (Fig 1).

### Methodology

The multi-phased workflow implemented in this investigation (Fig 2) is described comprehensively below.

### Preparation of location of COVID-19 active cases

A dataset of active cases of COVID-19 in Fars was prepared to analyze the relationships between the locations of active cases and the effective factors that may be useful for predicting outbreak risk. The data utilized in this research (S1 File) was collected on April 10, 2020 from Iranian's Ministry of Health and Medical Education (IMHME: http://ird.behdasht.gov.ir/).

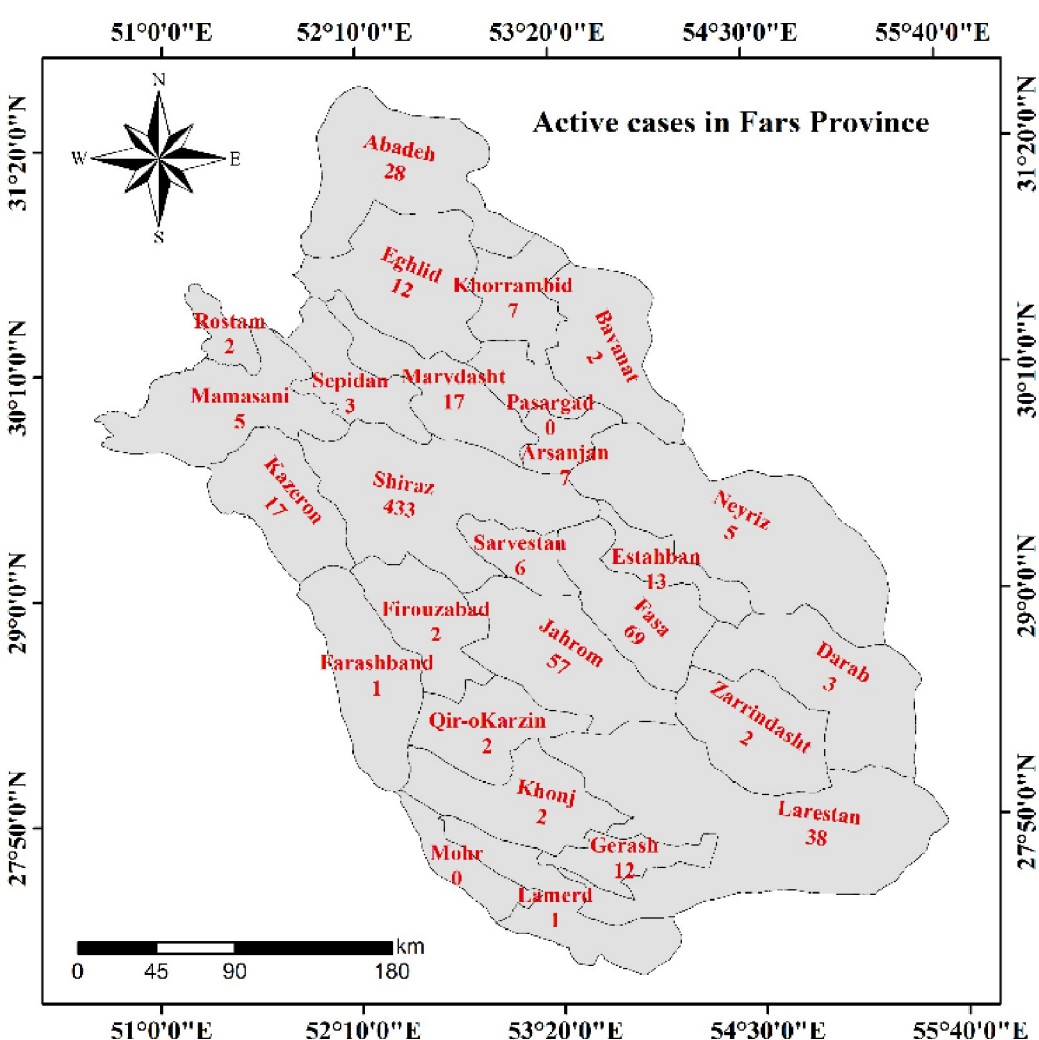

**Fig 1. The counties of Fars Province, Iran, and the number of COVID-19 infected case identified from March 29, 2020.**

## Preparation of effective factors

Choosing the appropriate effective factors to predict the risk of pandemic spread is vital as its quality affects the validity of the results [17]. Since, there have been no previous studies of risk for COVID-19 distribution, the selection of effective factors is a quite challenging task. Also, there is no approved universal factors for mapping the outbreak risk of COVID-19. Ongoing research on the pandemic has revealed that local and community-wide transmission of the virus largely happens in public places where the most people are likely to come into contact with largest number of potential carriers of the infection [26]. Wang et al. [27] indicated that meteorological conditions, such as rapidly warming temperatures in 439 cities around the world resulted in a decline of COVID-19 cases. Accordingly, in this research, we selected sixteen most relevant effective factors for the outbreak risk mapping of COVID-19 in Fars Province of Iran, which includes minimum temperature of coldest month (MTCM), maximum temperature of warmest month (MTWM), precipitation in wettest month (PWM), precipitation of driest month (PDM), distance from roads, distance from mosques, distance from hospitals, distance from fuel stations, human footprint, density of cities, distance from bus

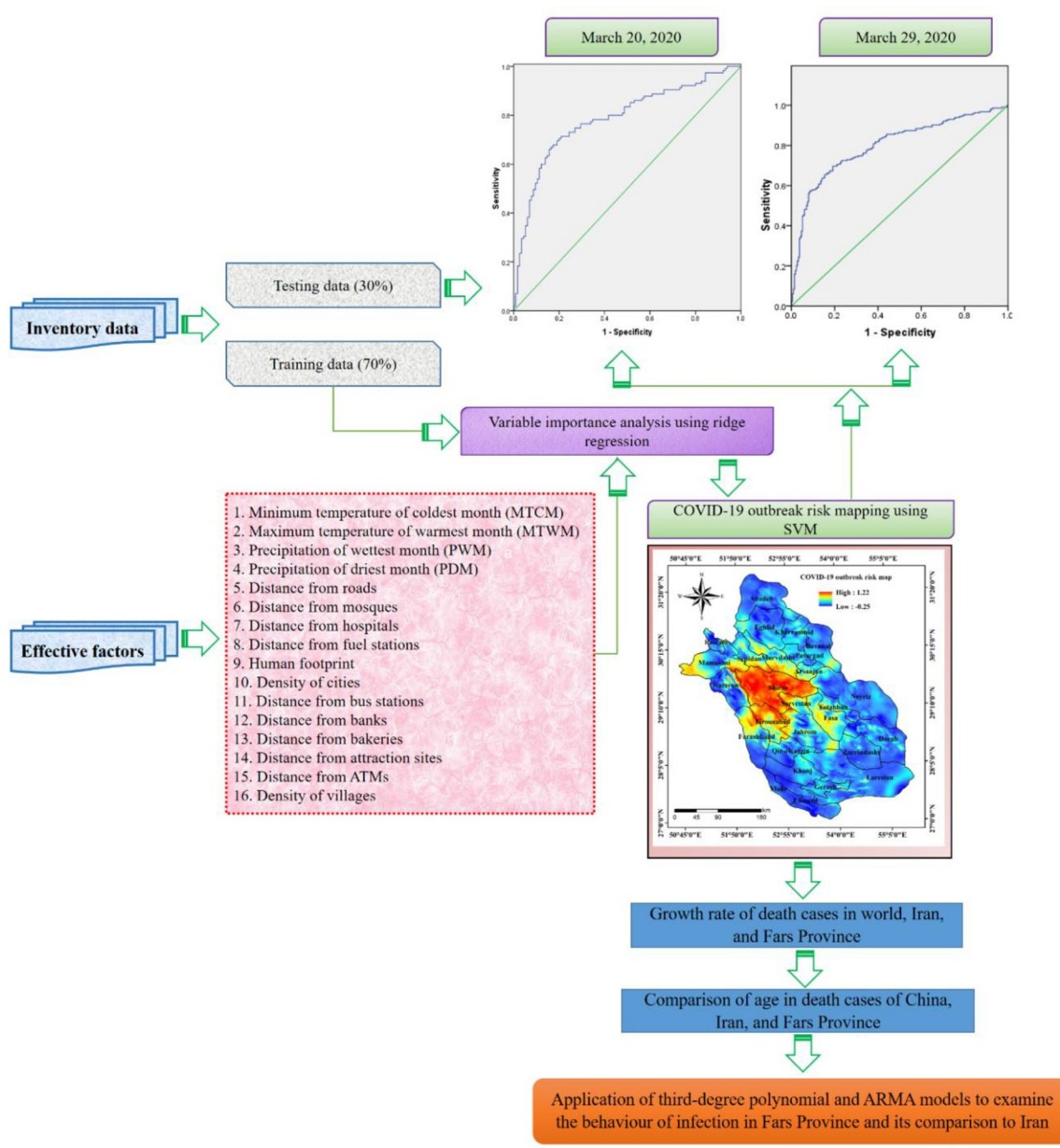

**Fig 2. The methodological framework followed in this study.**

stations, distance from banks, distance from bakeries, distance from attraction sites, distance from automated teller machines (ATMs) and density of villages. All the effective factors employed in this research are generated using ArcGIS 10.7.

A few studies have established that variation in temperature would impact the transmission of COVID-19 [27]. It has also been reported that alteration in temperature would have impacted the SARS outbreak, which was caused by the identical type of coronavirus as SARS-CoV-2 [28]. Recently, Ma et al. [2] disclosed that surge in temperature and humidity conditions have resulted in the decline of death caused by SARS-CoV-2. Thus, climatic factors such as temperature and precipitation can have an impact on the outbreak of SARS-CoV-2. The temperature and precipitation data, namely MTWM, MTCM, PDM and PCM of Fars

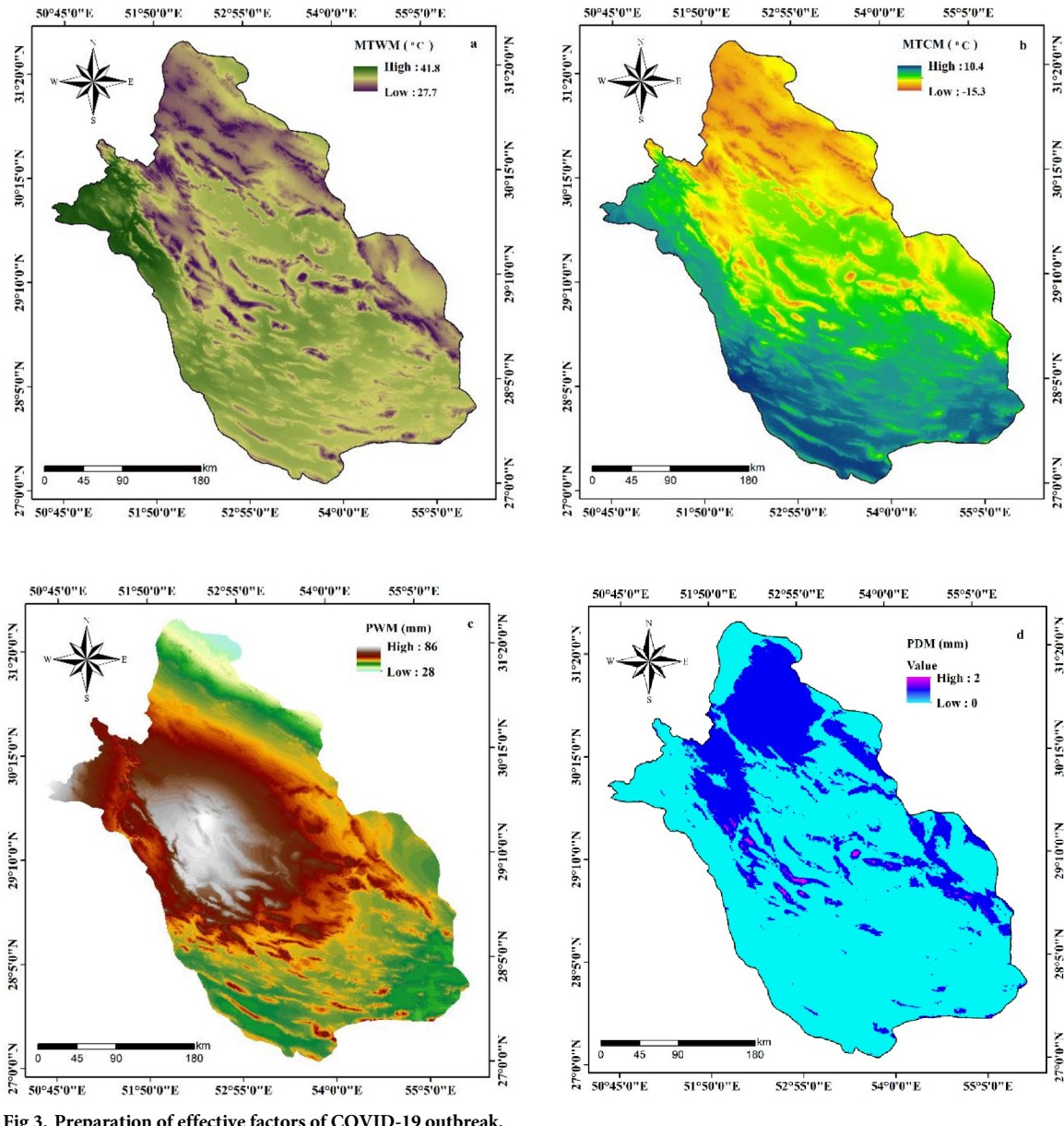

**Fig 3. Preparation of effective factors of COVID-19 outbreak.**

Province is acquired from world climatic data (https://www.worldclim.org/). In this study, the MTWM of the Fars Province ranges from 27.7˚C to 41.8˚C (Fig 3) whereas MTCM ranges between -15.3˚C and 10.4˚C (Fig 3). The PWM of the study area varies between 28 mm and 86 mm (Fig 4), and also the PDM is presented in Fig 3.

The proximity to various public places including roads, mosques, hospitals, fuel stations, bus stations, banks, bakeries, attraction sites, and ATMs where people come in close contact to each other can also be considered as significant factors that influence the distribution of COVID-19. The data was acquired from Open Street Map (https://www.openstreetmap.org). The distance from roads ranges from 0 to 45 in the study area (Fig 4) whereas the distance from mosques varies between 0 and 0.71 (Fig 4) and the distance from fuel stations spans 0 to 0.67 (Fig 4). The distance from bus stations, banks, bakeries, attraction sites, and ATMs of Fars

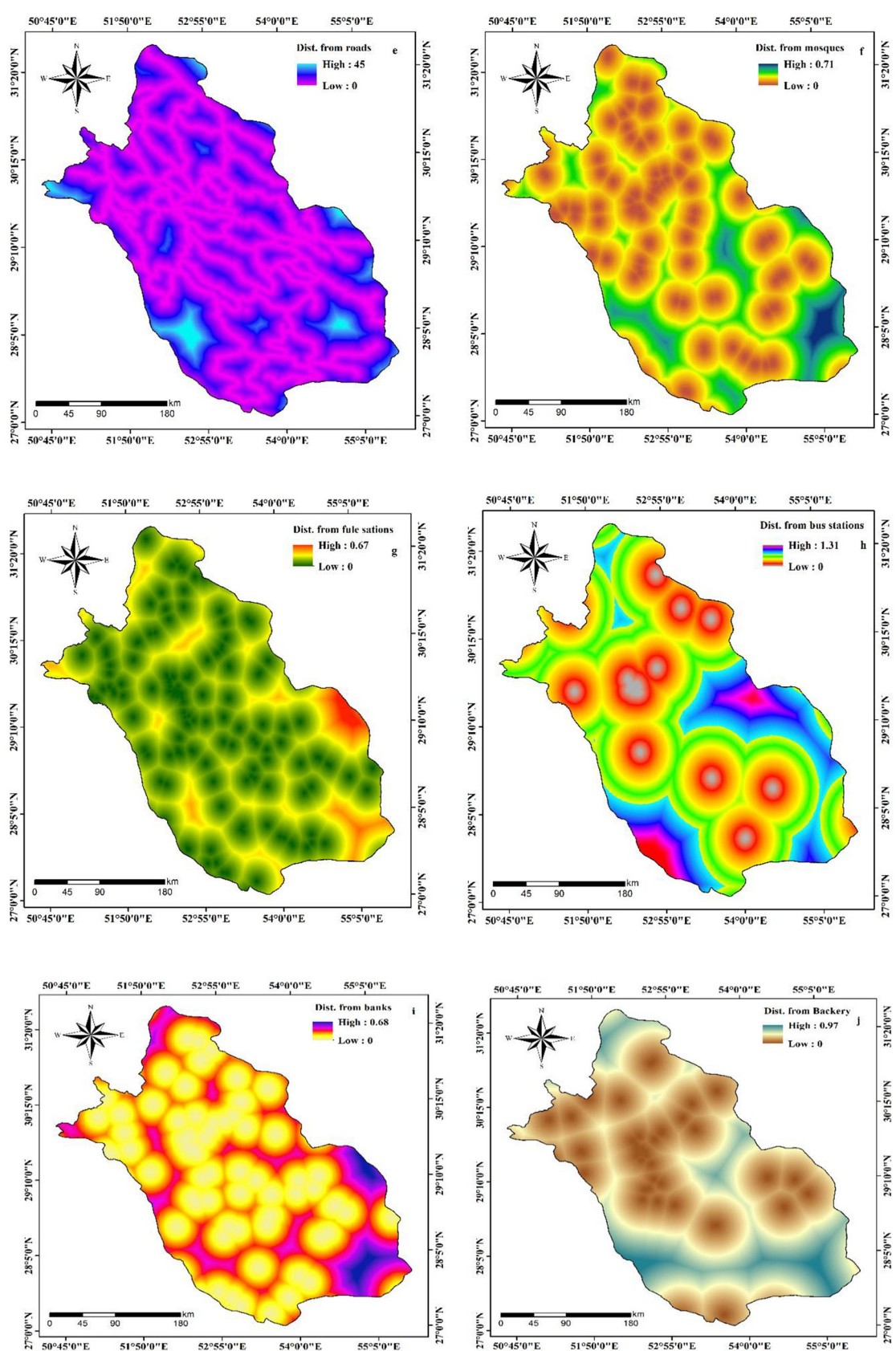

**Fig 4. Preparation of effective factors of COVID-19 outbreak.**

Province have the minimum value of 0 and maximum value of 1.31, 0.68, 0.97, 0.79, and 0.78 respectively (Figs 4, 5). Since humans are the potential carriers of the COVID-19, the use of human footprint (HFP) can aid in understanding the terrestrial biomes on which humans have more influence and access [29]. In this study, HFP of the study area is acquired from the Global Human Footprint Dataset. The HFP of Fars Province ranges from 6 to 78 (Fig 5) where the minimum value represents the places having least access by humans, and the maximum value refers to those regions having highest human influence and access. The density of population is also considered to be an important factor for the spread of the disease [30, 31]. Gilbert et al. [32] revealed that the number of COVID-19 cases was proportional to the population density in Africa. Accordingly, in this research, density of cities and villages were assessed, and the outcome displays that density of cities in Fars Province ranges between 0 and 0.60 (Fig 5) while the density of villages varies from 0 to 0.58 (Fig 5). The distance from hospitals ranged from 0 to 1.11 (Fig 5).

## Evaluation of variable importance using ridge regression

The association among the location of COVID-19 active cases and effective factors were evaluated using ridge regression in order to assess the significance of individual effective factor in predicting the outbreak risk [17]. To our knowledge, no previous study in epidemic outbreak risk mapping has utilized ridge regression in determining the significance of effective factors. However, the ridge regression algorithm has been utilized for modelling purposes in various fields [33]. It was first given by Hoerl and Kennard [34] which exploits $L_2$ norm of regularization for lessening the model complication and controlling overfitting. Ridge regression was also developed to avoid the excessive instability and collinearity problem caused by least-square estimator [35]. The 'caret' package (https://cran.r-project.org/web/packages/caret/caret.pdf) of R 3.5.3 was utilized for assessing the variable importance using ridge regression.

## Machine learning algorithm (MLA)

**Support vector machine.** SVM is an extensively exercised MLA in diverse fields of research that functions on the principle of statistical learning concept and structural risk minimization given by Vapnik [36], which is utilized for classification as well as regression intricacies [37, 38]. SVM has high efficacy in classifying both linearly separable and inseparable data classes [39]. It utilizes an optimal hyperplane to distinguish linearly divisible data, whereas kernel functions are employed for transforming inseparable data into a higher dimensional space so that it can be easily categorized [40]. Assume a calibration dataset to be $(s_m, t_m)$, where m is 1, 2, 3. . ., x; $s_m$ refers to the sixteen independent factors; $t_m$ denotes 0 and 1 that resembles risk and non-risk classes and x represents the total amount of calibration data. This algorithm tries to obtain an optimal hyperplane for classifying the aforementioned classes by utilizing the distance between them, which can be formulated as follows [41]:

$$\frac{1}{2}\|p\|^2 \qquad (1)$$

$$t_m((p \times s_m) + a) \geq 1 \qquad (2)$$

where, $\|p\|$ denotes the rule of normal hyperplane; a refers to a constant. When Lagrangian multiplier $(\lambda_m)$ and cost function is introduced, the expression can be given as follows [42]:

$$l = \frac{1}{2}\|p\|^2 - \sum_{n=1}^{x} \lambda_m(t_m((p \times s_m) + a) - 1) \qquad (3)$$

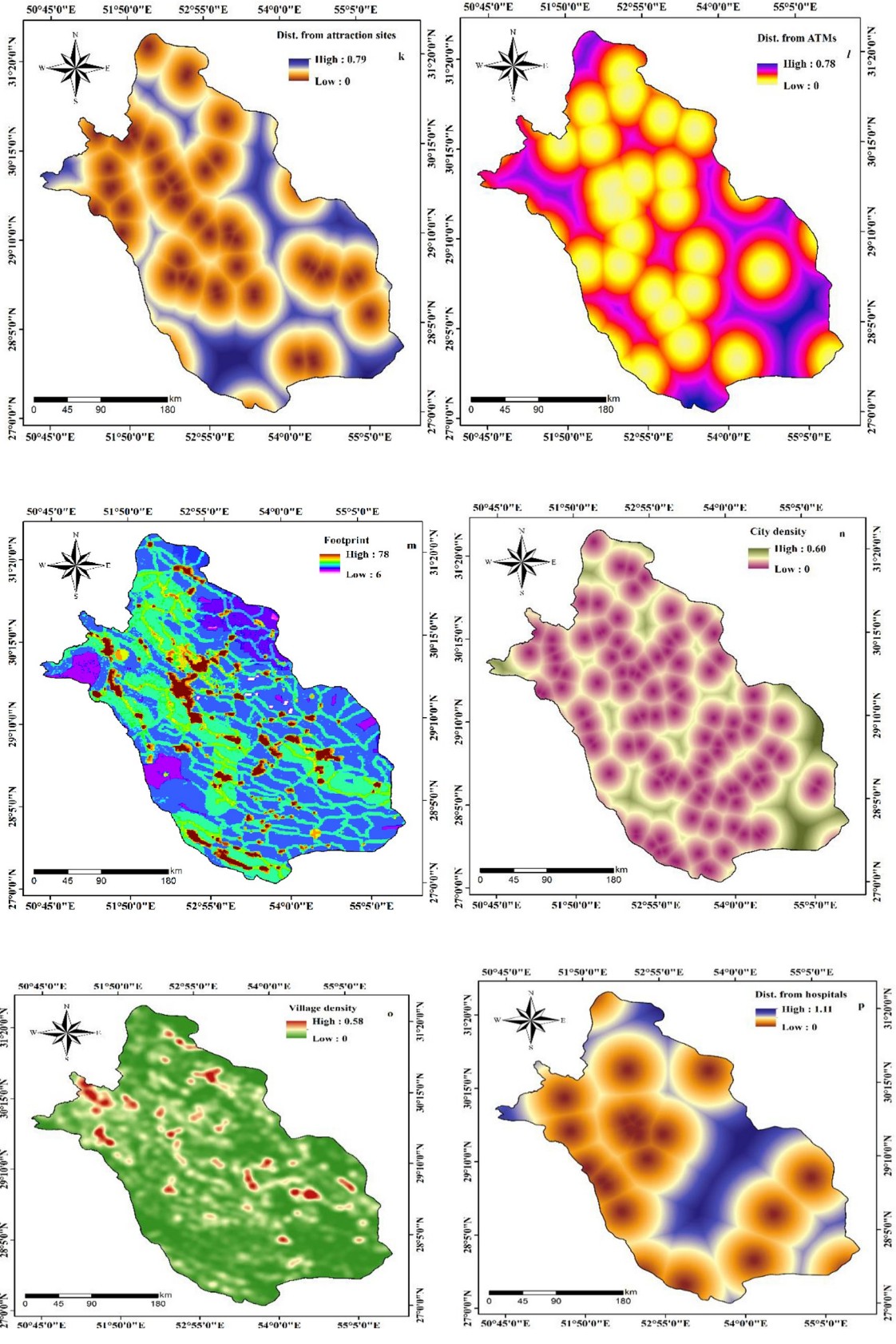

**Fig 5. Preparation of effective factors of COVID-19 outbreak.**

In case of an inseparable dataset, a slack covariate $\delta_m$ is added into the previs Eq (2) that is provided as follows [36]:

$$t_m((p \times s_m) + a) \geq 1 - \delta_m \qquad (4)$$

Accordingly Eq (3) can be described as follows [36]:

$$L = \frac{1}{2}\|p\|^2 - \frac{1}{ux}\sum_{n=1}^{x}\delta_m \qquad (5)$$

Moreover, SVM contains four kernel functions (linear, polynomial, radial basis function: RBF and sigmoid) for making an optimal margin in case of inseparable dataset [36]. Mohammadinia et al. [20] revealed that RBF kernel type produces high prediction accuracy than other kernel types for epidemic outbreak risk mapping. Thus, in this study, RBF is used for creating decision boundaries, and the kernel function is expressed as follows [36]:

$$K(z_a, z_b) = (-v\|z_a - z_b\|), v > 0 \qquad (6)$$

where, $K(z_a, z_b)$ refers to kernel function and $v$ represents its parameter.

**Analysis of growth rate for active and death cases of COVID-19.**   In this study, the growth rate (GR) of active and death cases around the world, Iran, and Fars Province were evaluated using the data acquired from WHO and IMHME between February 25, 2020 and June 10, 2020 for active cases and from March 2, 2020 to June 10, 2020 for death cases.

**Validation of outbreak risk map.**   The cross-checking of the calibrated model using untouched testing data is vital for determining the scientific robustness of the prediction [37]. In this research, we utilized ROC- AUC curve values for the validation of COVID-19 outbreak risk map generated using the SVM model. It is a widely utilized validation technique for analyzing the predictive ability of a model [39]. A model is determined to be perfect, very good, good, moderate and poor if the AUC values were 1.0–0.9, 0.9–0.8, 0.8–0.7, 0.7–0.6 and 0.6–0.5, respectively [43].

**Models for infection cases trend.**   The behavior of the variable infection cases in Fars province was captured by a third-degree polynomial or cubic specification while for those of Iran the fourth-degree polynomial specifications was found to be more appropriate as follows:

$$Infection(t) = \alpha_1 t + \alpha_2 t^2 + \alpha_3 t^3 + \alpha_4 t^4 \qquad (7)$$

where, $Infection(t)$ represents the total infected cases in day t and t denotes the days starting from 19th of February for Iran and one week later for Fars province. A quadratic specification was examined and based on the fitted model, the cubic form (for Fars province) and fourth-degree polynomial (for Iran) were selected. In the literature, the cubic form of specification has been applied by Aik et al. [44] to examine the Salmonellosis incidence in Singapore. We also used an ARMA model to compare the process generating the variable for Iran and Fars province. This model includes two processes: Autoregressive (AR) and Moving Average (MA) process. An ARMA model of order (p,q) can be written as [45]:

$$x(t) = \beta_0 + \sum_{i=1}^{p}\beta_i x_{t-i} + \sum_{j=1}^{q}\beta_j \varepsilon_{t-j} \qquad (8)$$

Where x is the dependent variable and $\varepsilon$ is the white noise stochastic error term. In the applied model, x shows the total infected cases and t is the days starting from the first day of happening infection cases. In building a time series model, the data are expected to be stationary [24]. In other words, the model (Eq 8) is based on the assumption that the data series are stationary. Briefly, a time series process $x(t)$ is stationary if the mean and variance are constant

and independent of time and the covariance between $x(t)$ and $x(t+s)$ ($x$ with $s$ period apart) is time-invariant or is dependent only upon the distance between the two time periods considered [46, 47]. Thus, if a time series have time-varying mean or a time-varying variance or both will be nonstationary. Using nonstationary time series for the forecasting purposes has little practical value. If the applied time series data is not stationary, after differencing it $d$ times an stationary time series was obtained. This series is called integrated of order $d$. After differencing $d$ times, we may apply the ARMA (p, q) model which is called ARIMA (p, d, q) [46]. The ARIMA (p,d,q) model is an ARMA(p,q) that applies $d$ times differencing data. Benvenuto et al. [23] applied an ARIMA model to predict the epidemiological trend of COVID-2019. Also, Saba and Elsheikhb [21] used this model to forecast the outbreak of COVID-19 in Egypt.

## Results

### Outcome of the variable importance analysis

The analysis of variable importance using ridge regression revealed that distance from bus stations, distance from hospitals, and distance from bakeries have the highest significance whereas distance from ATMs, distance from attraction sites, distance from fuel stations, distance from mosques, distance from road, MTCM, density of cities and density of villages exhibit moderate importance. The effective factors such as distance from banks, MTWM, HFP, PWM and PDM were the least influential factors (Fig 6).

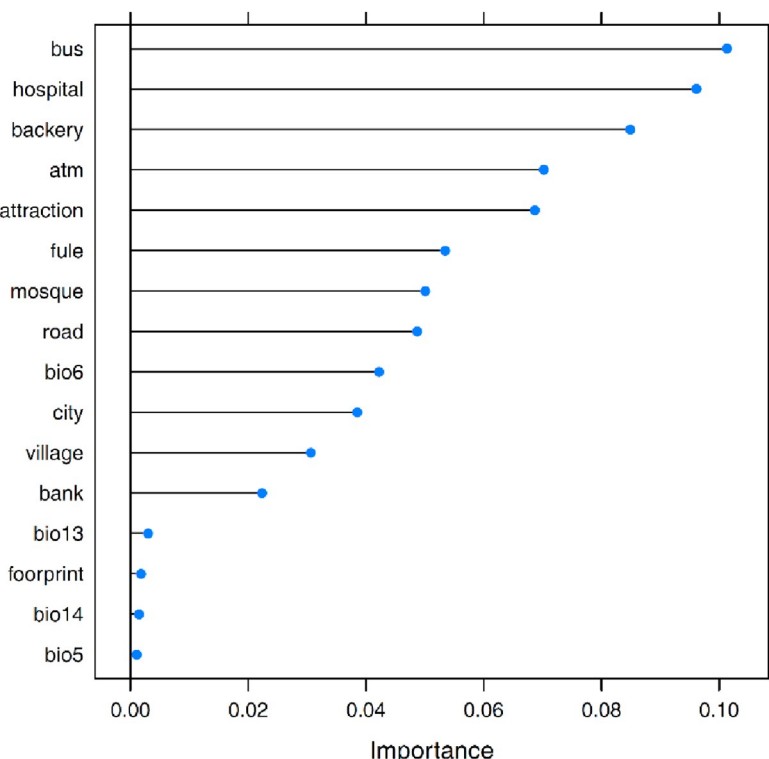

Fig 6. Variable importance of each effective factors (bus: Distance from bus stations; hospital: Distance from hospitals; bakery: Distance from bakeries; atm: Distance from ATMs; attraction: Distance from attraction sites; fuel: Distance from fuel stations; mosque: Distance from mosques; road: Distance from road; bio6: MTCM; city: Density of cities; village: Density of villages; bank: Distance from banks; bio13: MTWM; footprint: HFP, bio14: PWM; bio5: PDM).

## COVID-19 outbreak risk map using SVM

The COVID-19 outbreak risk map generated using SVM displays that risk of SARS-CoV-2 ranges from -0.25 to 1.22 (March 29) and -0.35 to 1.21 (April 10) where -0.25 and -0.35 represent the lower risk of SARS-CoV-2 outbreak and 1.22 and 1.21 indicates the regions of Fars Province which is likely to experience a higher risk of COVID-19 outbreak (Fig 7A and 7B). It can be observed from Fig 7B (April 10) that Shiraz County and its surrounding counties including Firouzabad, Jahrom, Sarvestan, Arsanjan, Marvdasht, Sepidan, Abadeh, Khorrambid, Rostam, Larestan and Kazeron of Fars Province has the highest risk of being the epicentre of SARS-CoV-2 outbreak. Apart from which counties like Eghlid, and Fasa also lie in the high risk zone.

## Outcome of growth rate analysis

The results of GR of active cases in the world, Iran, and Fars Province are presented in Fig 8. Our results displayed that the highest active cases in the world, Iran, and Fars Province were related to March 11 (GR = 1.59), Feb 26 (GR = 2.41), and March 15 (GR = 4.8), respectively. Also, the outcome stated that GR average of active cases in the world, Iran, and Fars Province reported since February 25 to June 10 was 1.15, 1.06, and 1.06, respectively. Our observations demonstrated that the highest GR of active cases in Fars Province was on March 16 (GR = 4.80), March 28 (GR = 4.10), March 09 (GR = 3.20), April 19 (GR = 3.15), March 20 (GR = 2.40), June 2nd (2.14), March 22 (GR = 2.10), April 1st (GR = 2.10), and February 26 (GR = 2.00). On the other hand, the analyses indicated that between February 27 and February 29, the GR of active cases was zero in Fars Province, followed by a GR value of 0.38 in 05 June, 0.3 in March 14, March 19, March 21, and 0.26 in April 18, whereas the lowest GR of active cases in world and Iran observed on April 26 (GR = 0.81) and March 3 (GR = 0.67) respectively.

Death cases in the world, Iran, and Fars Province are given in Fig 9.

In total of 7131 active cases of COVID-19 in Fars Province, 118 died between February 24 and June 10. The highest GR of death cases in Fars Province was reported on April 15 (GR = 5.00), April 11 (GR = 4.00), March 24 (GR = 4.00), April 20 (GR = 3.00), March 26 (GR = 3.00), March 22 (GR = 2.00), March 4 (GR = 2.00), April 4 (GR = 2.00), and June l0 (GR = 2.00). Our analyses showed that since March 5 to March 11, March 15 to March 21, March 28 to April 3, April 5 to April 7, May 2 to May 5, May 8 to May 18, May 20 to May 26, and May 29 to June 7, the GR of death cases was equal to zero. Although the deaths on March 31, April 3, April 7, April 10, April 18, April 23, May 5, May 18, May 21, May 26, June 3, and June 7 were 3, 2, 4, 1, 2, 4, 4, 2, 2, 2, 3 and 1, respectively, the daily growth rate is zero. Also, average of the GR in Fars Province during 102 days was 0.77, whereas this rate in world and Iran was observed as 1.07 and 1.05, respectively. Fig 9 shows that the highest GR of death cases in the world and Iran was nearly equal during March 08 (GR = 2.17) and March 04 (GR = 2.50). In contrast, the lowest rate of death case was observed on April 26 (GR = 0.62), and May 25 (GR = 0.59).

Results of active cases in 31 provinces of Iran country by March 25 is presented in Fig 10. Observations indicate that the number of active cases in the 100,000 population varies from 4.4 to 86.1. This figure also shows that provinces of Sistan and Baluchestan and Bushehr have the lowest cumulative rate of active cases, whereas the highest rate was observed in Qom, Semnan, Markazi, and Yazd. The Qom Province was the first place in Iran where the outbreak of COVID-19 was recorded. The latest news reported by the Iranian's Ministry of Health and Medical Education (IMHME) on June 10 determines the number of active cases of Fars province in the 100,000 population is 146.99 while this number has been 10.4 on March 25.

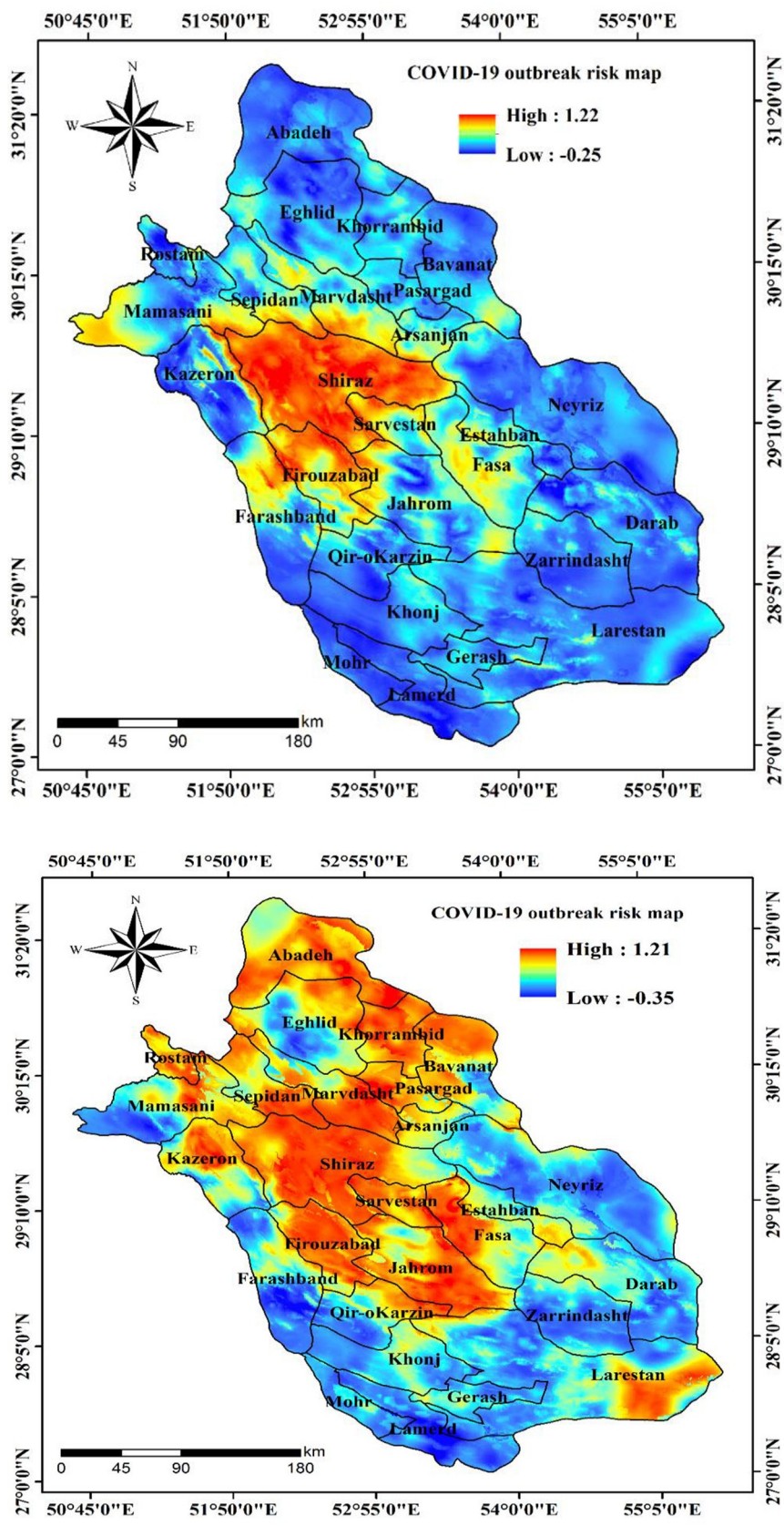

**Fig 7.** The COVID-19 outbreak risk map a) on March 29, 2020 and b) on April 10, 2020.

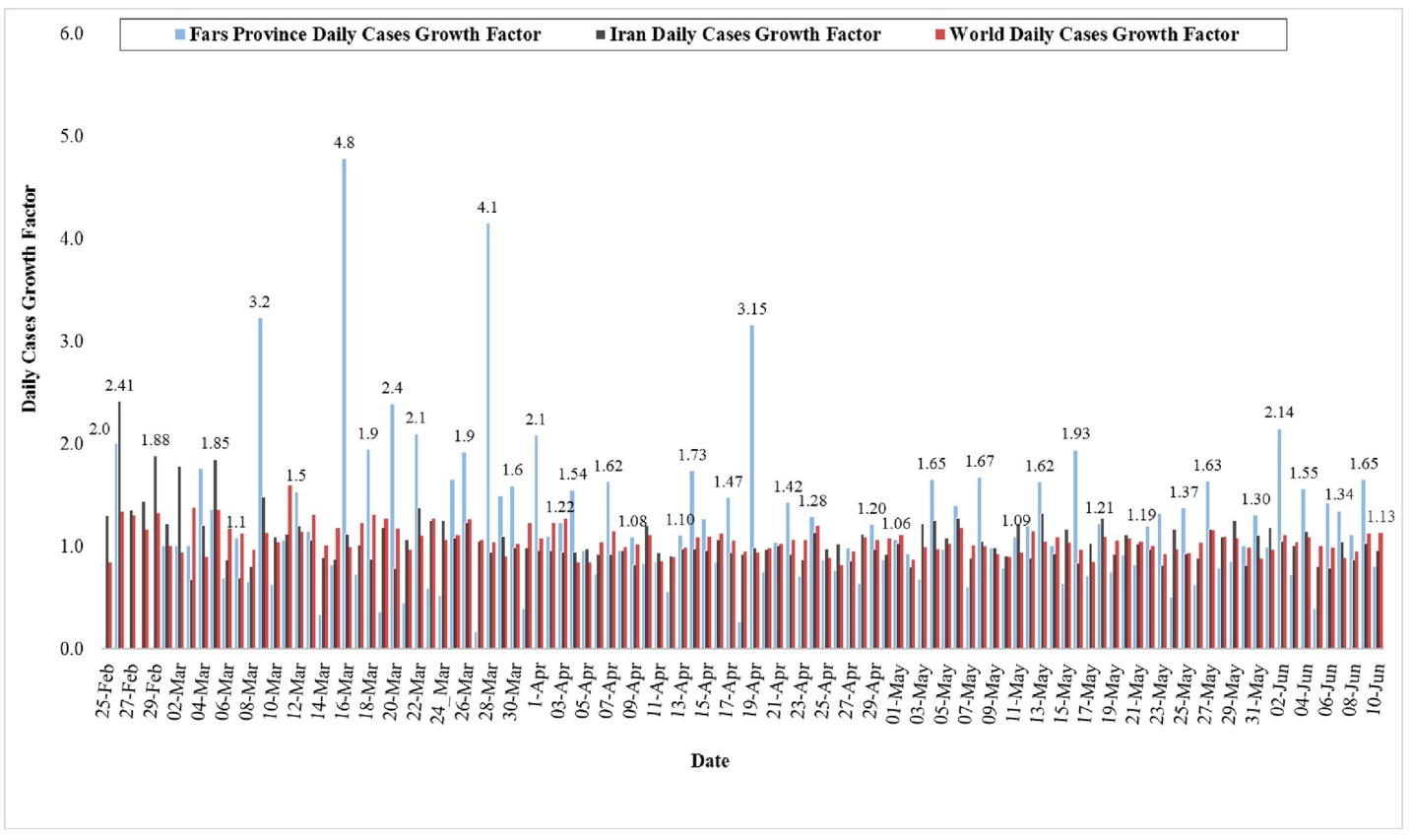

**Fig 8. Growth rate of active cases in the world, Iran, and Fars Province (From 25 February to 10 June 2020).**

A comparison among age class of death cases in China, Iran, and Fars Province is presented in Table 1. Percentage of death cases in China was related to February 29, whereas for Iran and Fars Province it is related to March 14 and May 4, respectively. Table 1 show that age class > 50 years old lie in the highest class of death rate. So, this age class of above 50 years is highly sensitive to COVID-19.

## Validation outcome of outbreak risk map

The ROC-AUC curve cross-validation technique is utilized in this research for validating the COVID-19 outbreak risk map generated by SVM. The model achieved an AUC value of 0.786 and a standard error of 0.031 indicating a good predictive accuracy when cross-verified using the remaining 30% testing dataset collected on March 20, 2020 (Fig 11 and Table 2).

When tested with active case locations on March 29, 2020, the model achieved an increased AUC value of 0.799 which proves the stable and good forecast precision of the outbreak risk map (Fig 12 and Table 3). Also, change detection on April 10, 2020 show that accuracy of the built models is increased to 86.6% (AUC = 0.868) (Fig 13 and Table 4).

## Comparison of Fars province and Iran infection cases

Two tools have been applied to compare the general trend of infection in Fars province and Iran. The first includes a third-degree (for Fars province) and a fourth-degree (for Iran) polynomial models that are presented in Fig 14. Another quantitative model is an ARIMA model

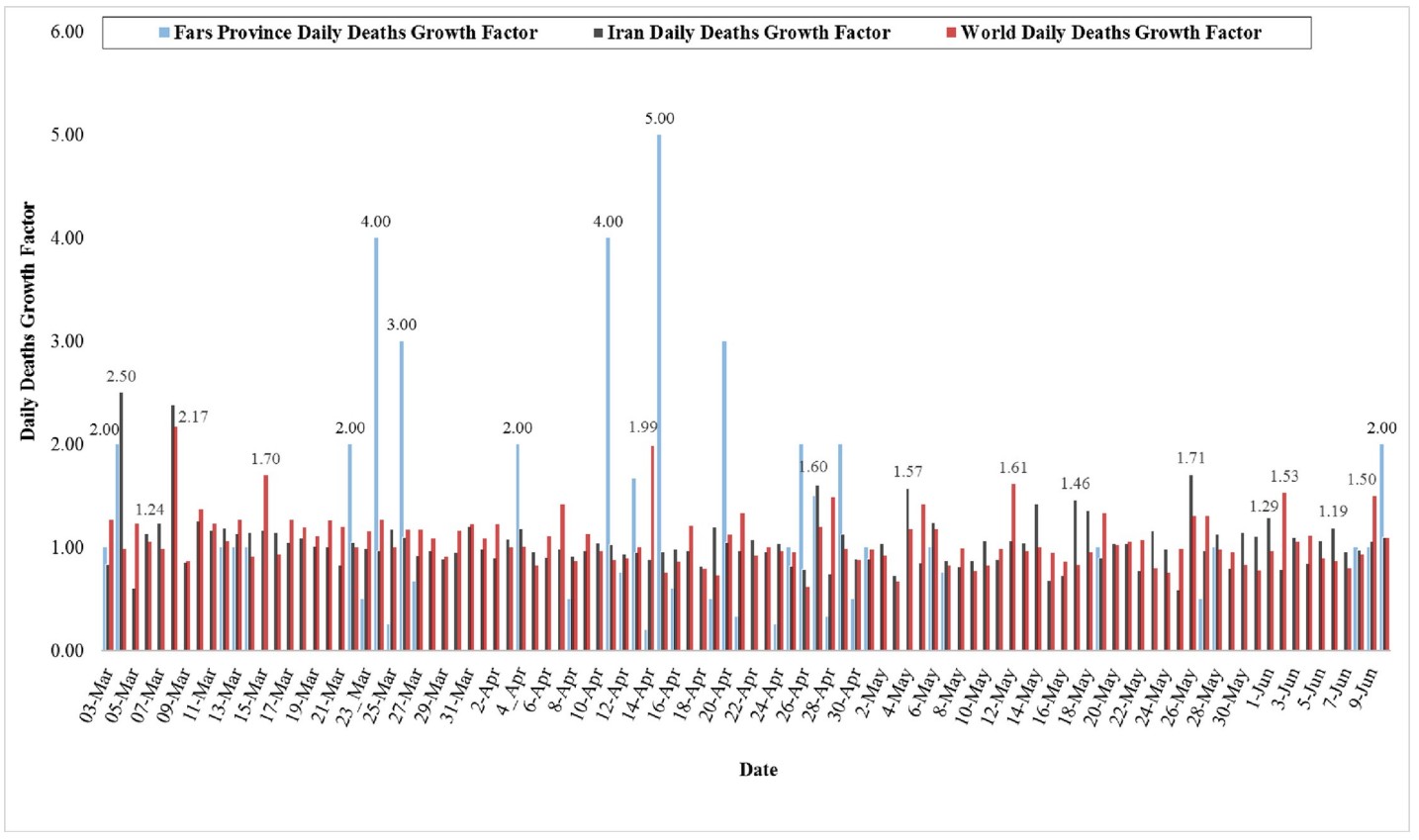

**Fig 9. Growth rate of death cases in the world, Iran, and Fars Province (From 2 March to 10 June 2020).**

presented in Table 5. Fig 12 shows the trend of infection cases in Iran and Fars province, where predicted values extraordinarily keep pace with the actual values. Coefficients of determination () values also indicate that estimated models have significant predictive power. The infection cases are increasing over the selected horizon.

The first derivative of the estimated model represents the daily infection cases. Based on the daily infection model, there is a turning point for both Iran and provincial cases. It was found that the turning point for provincial daily infection is 134. In other words, after 134 days the decreasing trend in the daily infection is expected.

However, the corresponding value for Iran is much higher than the provincial one. There are some evidences showing that a turning point in infection is expected. For instance, it has been reported for SARS incidence [48], HAV [49], ARI [50], and for A (H1N1)v. It is worth noting that a turning point means that after passing the peak, it is expected to show a decreasing trend. In the 107[th] day of infection, Fars province accounts for around 4.34% of the total Iranian cases while its population share is more than 6% (Statistical Center of Iran, 2016). Regarding the values obtained for turning points and the infection share, up to the present, the measures taken by the provincial government may be considered more effective than those taken in other provinces as a whole. However, it should be taken into consideration that Fars province experienced its first infection cases 7 days after Qom and Tehran, provinces that are considered as starting point for virus outbreak in Iran. This might have given the provincial governmental body and the households to take measures to cope with the widespread outbreak. It is worth noting that the comparison of the specified models is more appropriate to

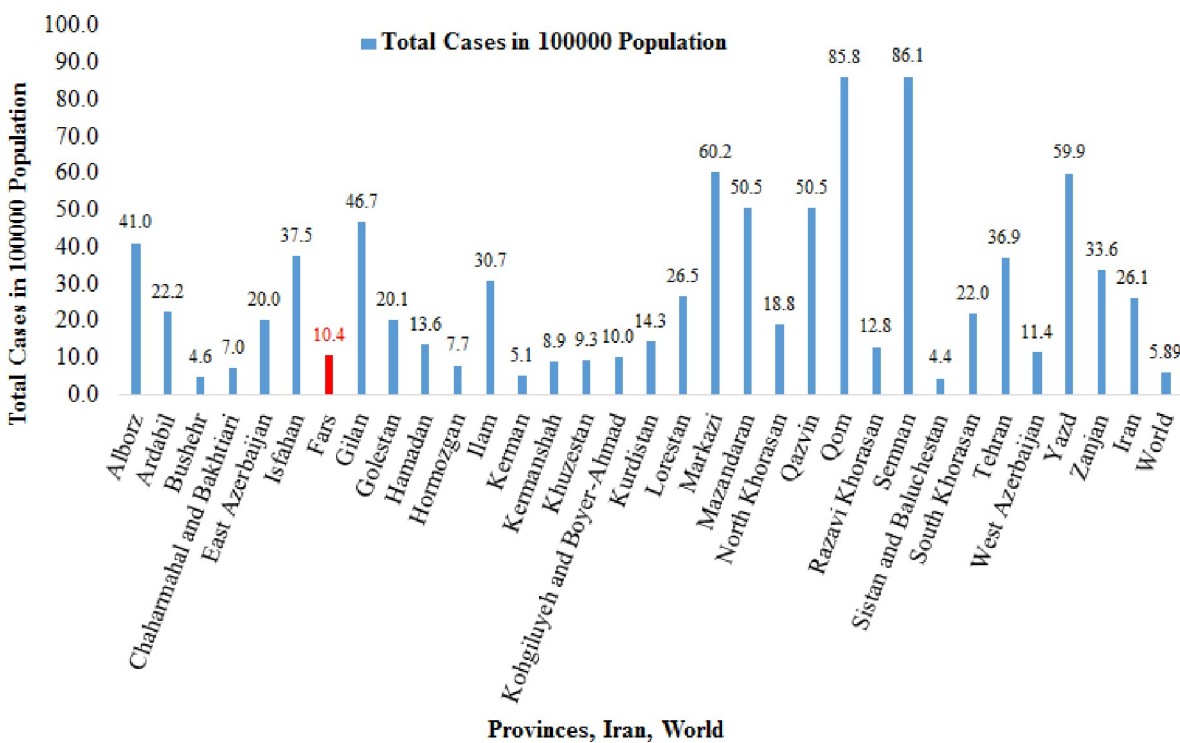

**Fig 10. Results of active cases in 31 provinces of Iran country by March 25, 2020.**

investigate the effectiveness of the measures taken by the corresponding health body rather than using it to predict future values.

The ARIMA time series models for infection variables of the Fars province and Iran are presented in Table 5. These models may show the generating process of the variables in time horizon. It is worth noting that in order to have more comparable models, a 107-day time horizon is selected. This is the period of time that data are available, starting on 19th of February for Iran and one week later for Fars province. As shown in Table 5, provincial data are generated by an ARIMA (2,1,1) process while ARIMA (2,0,2) was found more appropriate for Iran's data. Regarding the orders for AR and MA processes, the country model shows more complicated behaviour. In addition, the Fars data was applied after differencing since it was not stationary; indicating a more explosive process of an increasing trend for Fars province compared to those of Iran in the following days. The provincial data indicated more volatility which was captured by variance-related variable GARCH that was not easily captured in the trends as shown in Fig 14. Benvenuto et al. [23] also used an ARIMA model and found that COVID-2019 spread tends to reveal a slightly decreasing spread. Generally speaking, the diagnostic statistics indicate that the estimated models are acceptable since Q-statistics reveal that

**Table 1. Comparison of age in death cases of China, Iran, and Fars Province.**

| Country | China | Iran | Fars Province |
|---|---|---|---|
| Age | Death Rate (%) | Death Rate (%) | Death Rate (%) |
| >50 years old | 93.7 | 84.15 | 80 |
| 10–50 years old | 6.3 | 15.46 | 20 |
| <10 years old | 0 | 0.39 | 0 |

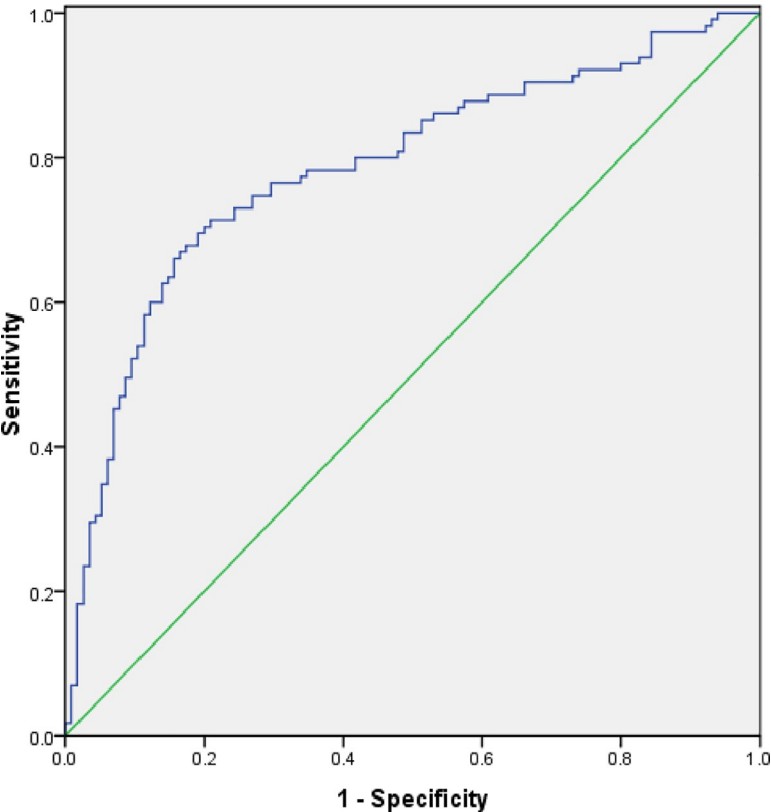

**Fig 11. Receiver operator characteristic (ROC) curve based on testing data from March 20, 2020.**

the residuals are not significantly correlated and the Jarque Berra statistic supports the normality of residuals at conventional significance level. In addition, all AR and MA roots were found to lie inside the unit circle, indicating that ARIMA process is (covariance) stationary and invertible.

## Discussion

There is a great necessity for new robust scientific outcomes that could aid in containing and preventing the COVID-19 pandemic from spreading. The spatial mapping of COVID-19 outbreak risk can aid governments and policy-makers in implementing strict measures in certain regions of a city or a country where the risk of an outbreak is very high. It is, therefore crucial to identify the regions that would have high outbreak risk through predictive modelling with the help of machine learning algorithms (MLAs). In recent times, MLAs have demonstrated promising results in forecasting the epidemic outbreak risk [17]. In this research, the SVM model showing good forecast accuracy was used for mapping the outbreak risk of COVID-19. Similarly, Mohammadinia et al. [20] revealed that GWR and SVM had the highest precision in mapping the occurrence of leptospirosis. Ding et al. [51] employed three MLAs including

**Table 2. Area under the curve based on data from March 20, 2020.**

| Area | Standard Error | Asymptotic Significant | Asymptotic 95% Confidence Interval | |
|---|---|---|---|---|
| | | | Lower Bound | Upper Bound |
| 0.786 | 0.031 | 0.000 | 0.726 | 0.846 |

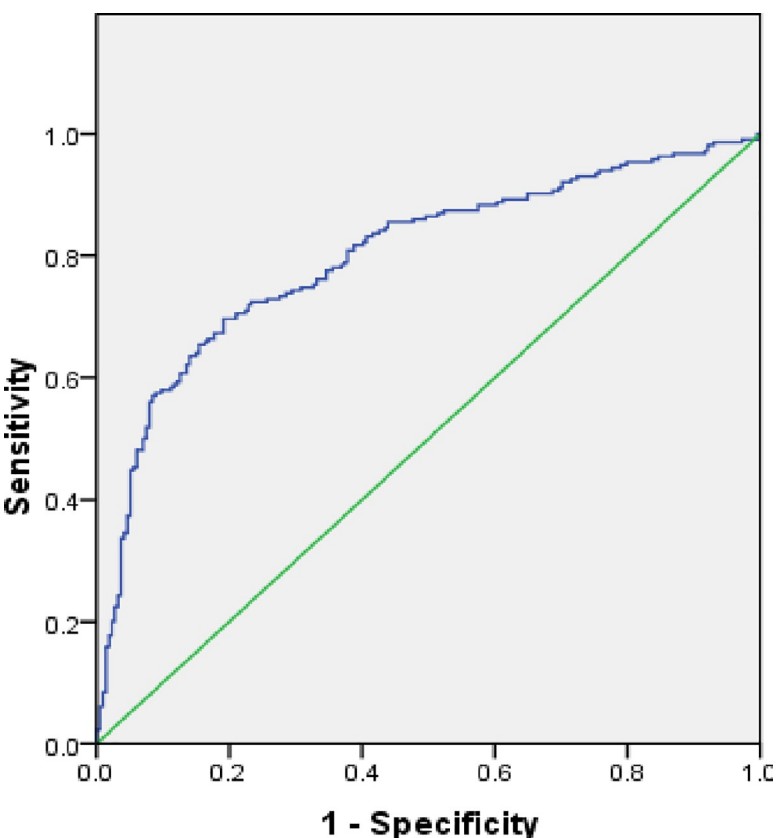

**Fig 12. Receiver operator characteristic (ROC) curve based on data from March 29, 2020.**

SVM, RF and GBM, for mapping the transmission risk assessment of mosquito-borne diseases and disclosed that all three MLAs acquired excellent validation outcome. Machado et al. [52] also applied RF, SVM and GBM in modelling the porcine epidemic diarrhoea virus and demonstrated 90% specificity values in case of SVM. Tien Bui et al. [17] stated that SVM achieved an AUC value of 0.968 in mapping the susceptibility to malaria. The ability to classify inseparable data classes is the greatest benefit of the SVM model [53]. It is among the most precise and robust MLA [54]. SVM can be useful and has higher prediction accuracy when it comes to handling a small dataset. However, Huang and Zhao [55] demonstrated that SVM also yields excellent precision in predictive modelling when a large dataset is utilized. The algorithm has a very low probability of overfitting and is not disproportionately impacted by noisy data [53]. Behzad et al. [56] revealed that SVM had huge capacity in simplification and had enduring forecast accuracy. It should also be noted that the predictive exactness of SVM model largely depends on the choice of kernel function [54]. Among the four kernel functions of SVM, RBF has been proved to generate high accuracy models [54]. SVM includes diverse kinds of categorization functions which are responsible for assessing overfitting and simplifying data that

**Table 3. Area under the curve based on data from March 29, 2020.**

| Area | Standard Error | Asymptotic Significant | Asymptotic 95% Confidence Interval | |
|---|---|---|---|---|
| | | | Lower Bound | Upper Bound |
| 0.799 | 0.022 | 0.000 | 0.756 | 0.841 |

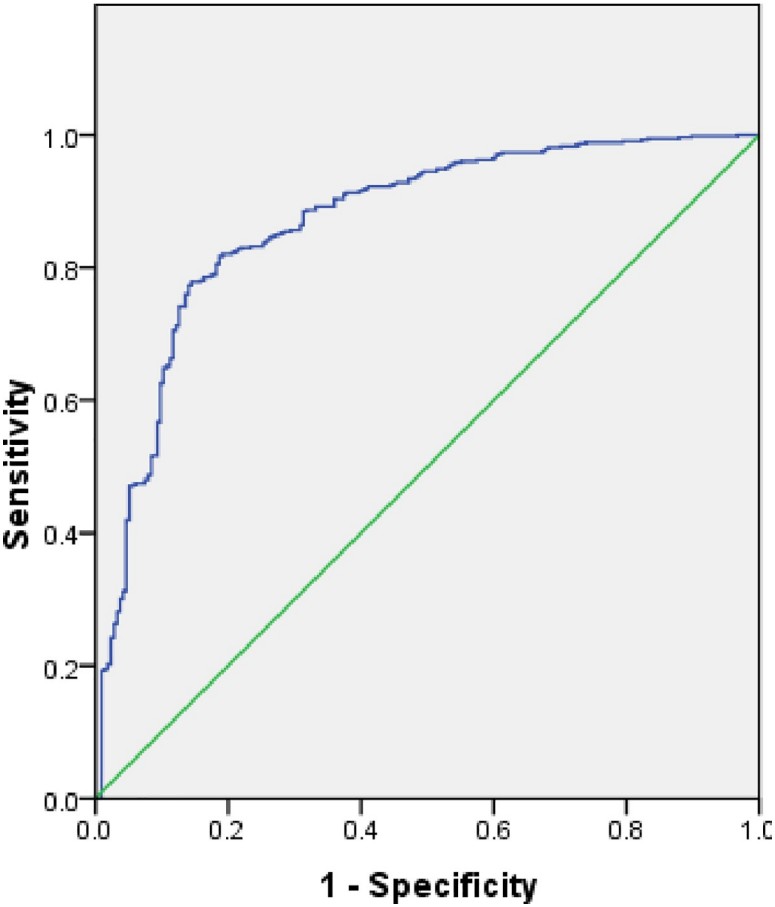

**Fig 13. Receiver operator characteristic (ROC) curve based on data from April 10, 2020.**

needs a minor tuning of model parameters [57]. The significance of each effective factor employed in this research is assessed using ridge regression. Since, there is no previous study in COVID-19 that outlines the proper effective factors. The outcome of this research can be very helpful for scientists in experimenting the same and additional effective factors for COVID-19 outbreak risk mapping. The proximity factors including distance from bus stations, distance from hospitals, distance from bakeries were most influential in forecasting the COVID-19 outbreak risk whereas other proximity factors such as distance from ATMs, distance from attraction sites, distance from fuel stations, distance from mosques and distance from road had the moderate influence which is followed by MTCM, density of cities and density of villages. It should be noted that climatic factors including MTWM, PWM and PDM

**Table 4. Area under the curve based on data from April 10, 2020.**

| Area | Standard Error | Asymptotic Significant | Asymptotic 95% Confidence Interval | |
|---|---|---|---|---|
| | | | Lower Bound | Upper Bound |
| .868 | .015 | .000 | .838 | .898 |

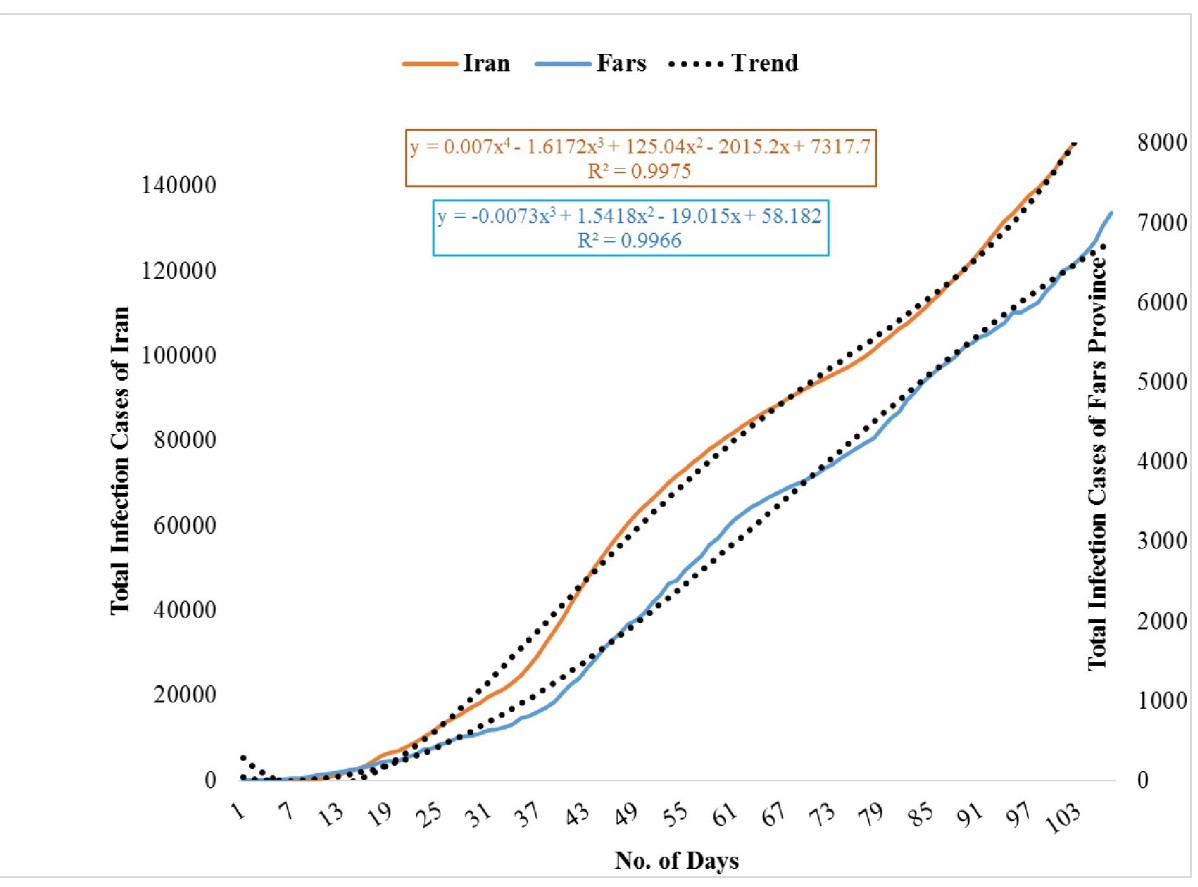

**Fig 14. Actual cases versus estimated cases in Fars province and Iran (From 25 February to 10 June 2020).**

had the least significance in mapping the outbreak risk. From this, it can be concluded that precipitation factors PWM and PDM are not associated with the transmission of COVID-19 in Fars Province whereas in case of temperature factors MTCM had moderate influence in mapping COVID-19 outbreak risk but MTWM exhibited a least significance. This outcome reveals that proximity factors had high influence in the transmission of SARS-CoV-2. In addition, the study conducted disclosed that increase in temperature will not decline the SARS--CoV-2 cases, although it has been also revealed that increase in temperature and absolute humidity could decrease the death of patients affected by COVID-19 [58]. The polynomial and ARIMA models were applied to examine the behaviour of infection in Fars province and Iran. The general trend of infection in Iran and Fars province are similar while more volatility for provincial cases is expected. The methodology and effective factors used in this research can be adapted in studies investigated in other parts of the world for preventing and controlling the outbreak risk of COVID-19.

## Conclusions

Mapping of SARS-CoV-2 outbreak risk can aid decision-makers in drafting effective policies to minimize the spread of the disease. In this research, GIS-based SVM was used for mapping the COVID-19 outbreak risk in Fars Province of Iran. Sixteen effective factors including MTCM, MTWM, PWM, PDM, distance from roads, distance from mosques, distance from hospitals, distance from fuel stations, human footprint, density of cities, distance from bus

**Table 5. The results of autoregressive integrated moving average (ARIMA) model for COVID-19 infection cases of Fars province and Iran.**

|  | Regressor | Coefficient | Standard error | t-statistics | probability |
|---|---|---|---|---|---|
| Iran | Constant | 151503.8 | 95854.82 | 1.58 | 0.117 |
|  | AR(1) | 1.494 | 0.006 | 248.53 | 0.000 |
|  | AR(3) | -0.495 | 0.006 | -84.16 | 0.000 |
|  | MA(1) | 0.403 | 0.080 | 5.00 | 0.000 |
|  | MA(3) | 0.295 | 0.083 | 3.51 | 0.000 |
|  | Adjusted | 0.999 |  |  |  |
|  | Q(1)a | 2.328 |  |  | 0.127 |
|  | Q(2)a | 3.176 |  |  | 0.204 |
|  | Jarque Berra | 0.666 |  |  | 0.716 |
|  | Inverted AR roots | -0.50 |  |  |  |
|  | Inverted Ma roots | -0.83 |  |  |  |
| Fars province | Constant | 29.467 | 272.211 | 0.108 | 0.913 |
|  | AR(1) | 0.425 | 0.096 | 4.42 | 0.000 |
|  | AR(2) | 0.554 | 0.107 | 5.18 | 0.000 |
|  | MA(5) | 0.214 | 0.109 | 1.95 | 0.050 |
|  | GARCH(-1) | -0.964 | 0.028 | -34.12 | 0.000 |
|  | Adjusted | 0.645 |  |  |  |
|  | Q(1)a | 3.147 |  |  | 0.076 |
|  | Q(2)a | 3.302 |  |  | 0.192 |
|  | Jarque Berra | 4.017 |  |  | 0.134 |
|  | Inverted AR roots | 0.99 |  |  |  |
|  | Inverted MA roots | -0.73 |  |  |  |

a $Q(P)$ is the significance level of the Ljung–Box statistics in which the first p of the residual autocorrelations are jointly equal to zero.

stations, distance from banks, distance from bakeries, distance from attraction sites, distance from automated teller machines (ATMs) and density of villages were selected along with the locations of active cases of SARS-CoV-2. The results of ridge regression revealed that distance from bus stations, distance from hospitals, and distance from bakeries had the highest influence in COVID-19 outbreak risk mapping whereas the climatic factors had the lowest influence. The generated model using SVM had a good predictive accuracy of 0.786 and 0.799 when verified with the locations of active cases during March 20 and March 29, 2020. However, the weakness of the SVM model lies in managing a very large dataset and inferring with the model outcome that is due to the black box nature of the model. The GR average for active cases in Fars for a period of 107 days was 1.15, whilst it was 1.06 in the country and the world. The Iranian government should take restrict preventive measures for controlling the outbreak of SARS-CoV-2 in Shiraz as a tourism destination and the counties having high risk. Based on the results of polynomial and an ARIMA model, the infection behavior is not expected to reveal an explosive process, however; the general trend of infection will last for several months especially in Iran as a whole. A more slowly trend is expected in Fars Province, demonstrating extensive home quarantine and travel and movement restrictions were good strategies for disease control in Fars province. The main policy implication is that the infection cases, to some extent, may be controlled using more effective measures. Although, the estimated models may be used to predict the infection in following days, however; this contribution is less significant than the other implications derived from them. Generally speaking, it is expected to encounter a decreasing trend, however; this may be reversed if the ongoing attempts are slowed down, pointing out the need to keep the measures like quarantine or even to try more restricting

attempts. As a policy implication, it is worth noting that the applied models clearly show the extent that the measures taken by the central and provincial governments body have been efficient, allowing them to consider more effective measures. This contribution will be more valuable when the dynamic and the complicated nature of the virus is taken into consideration. Several extensions may be recommended for further investigation. It is possible to apply the developed models to examine the behaviour of other related variables including recovered cases and critical cases. If more detailed data is provided, the effectiveness of the location-specific measures deserves to be investigated more deeply.

## Supporting information

**S1 File.**
(XLSX)

## Author Contributions

**Conceptualization:** Hamid Reza Pourghasemi, Bahram Heidari.

**Data curation:** Hamid Reza Pourghasemi, Soheila Pouyan, Bahram Heidari.

**Formal analysis:** Hamid Reza Pourghasemi, Soheila Pouyan, Zakariya Farajzadeh, Bahram Heidari.

**Funding acquisition:** Hamid Reza Pourghasemi.

**Investigation:** Hamid Reza Pourghasemi, Soheila Pouyan, Zakariya Farajzadeh, Bahram Heidari.

**Methodology:** Hamid Reza Pourghasemi, Soheila Pouyan.

**Project administration:** Hamid Reza Pourghasemi.

**Resources:** Hamid Reza Pourghasemi.

**Software:** Hamid Reza Pourghasemi, Soheila Pouyan, Zakariya Farajzadeh, Sedigheh Babaei.

**Supervision:** Hamid Reza Pourghasemi, Bahram Heidari, John P. Tiefenbacher.

**Validation:** Hamid Reza Pourghasemi, Zakariya Farajzadeh.

**Visualization:** Hamid Reza Pourghasemi, Nitheshnirmal Sadhasivam, Sedigheh Babaei, John P. Tiefenbacher.

**Writing – original draft:** Hamid Reza Pourghasemi, Soheila Pouyan, Bahram Heidari.

**Writing – review & editing:** Hamid Reza Pourghasemi, Zakariya Farajzadeh, Nitheshnirmal Sadhasivam, Bahram Heidari, Sedigheh Babaei, John P. Tiefenbacher.

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
