## [Decision Letter · Decision Letter 0]

15 Jun 2020

PONE-D-20-10865

Assessment of the outbreak risk, mapping and infestation behavior of COVID-19: Application of the autoregressive and moving average (ARMA) and polynomial models

PLOS ONE

Dear Dr. Heidari,

Thank you for submitting your manuscript to PLOS ONE. After careful consideration, we feel that it has merit but does not fully meet PLOS ONE’s publication criteria as it currently stands. Therefore, we invite you to submit a revised version of the manuscript that addresses the points raised during the review process.

Please revise the paper by considering the reviewers' comments.

We look forward to receiving your revised manuscript.

Kind regards,

Jie Zhang

Academic Editor

PLOS ONE

Journal Requirements:

2. "PLOS specifies that experiments, statistics, and other analyses are performed to a high technical standard; sample sizes are large enough to produce robust results; and methods are described in sufficient detail to allow another researcher to reproduce the experiment (http://journals.plos.org/plosone/s/criteria-for-publication#loc-3). As such, we ask you to describe where to access the Iranian’s Ministry of Health and Medical Education data used in the study. Please provide a link or include the relevant data as a supplementary file.

3.  We note that Figures 1, 2, 3 and 5 in your submission contain map images which may be copyrighted. All PLOS content is published under the Creative Commons Attribution License (CC BY 4.0), which means that the manuscript, images, and Supporting Information files will be freely available online, and any third party is permitted to access, download, copy, distribute, and use these materials in any way, even commercially, with proper attribution. For these reasons, we cannot publish previously copyrighted maps or satellite images created using proprietary data, such as Google software (Google Maps, Street View, and Earth). For more information, see our copyright guidelines: http://journals.plos.org/plosone/s/licenses-and-copyright.

a) You may seek permission from the original copyright holder of Figure(s) [#] to publish the content specifically under the CC BY 4.0 license. 

Reviewers' comments:

Reviewer's Responses to Questions

**Comments to the Author**

1. Is the manuscript technically sound, and do the data support the conclusions?

Reviewer #1: Partly

Reviewer #2: Yes

Reviewer #3: Yes

2. Has the statistical analysis been performed appropriately and rigorously? 

Reviewer #1: Yes

Reviewer #2: Yes

Reviewer #3: Yes

3. Have the authors made all data underlying the findings in their manuscript fully available?

Reviewer #1: Yes

Reviewer #2: Yes

Reviewer #3: Yes

4. Is the manuscript presented in an intelligible fashion and written in standard English?

Reviewer #1: No

Reviewer #2: Yes

Reviewer #3: Yes

5. Review Comments to the Author

Reviewer #1: This paper presents an interesting model for assessing the growth of covid-19.

In order to improve the paper's quality, some changes should be made.

1) Make the abstract clearer and more concise, both in relation to the approach adopted, justification, results achieved, and implications for monitoring the pandemic of COVID-19.

2) In the introduction, make clear to the reader the original objectives and contributions of the paper.

3) Justify the use of parameters in your approach method.

4) ) Highlight in the conclusion the results obtained, as well as the weaknesses of the adopted approach (computational, practical, ...)

Reviewer #2: 1- What are the advantages and the limitations of the proposed models?

2- How will this study help Iranian decision makers to develop their plans?

3- why does the death rate in Iran change with time, explain the trend?

4- Use more statistical criteria to evaluate the accuracy of the model, you may check the following paper

"An enhanced productivity prediction model of active solar still using artificial neural network and Harris Hawks optimizer"

5-You can strength you introduction using the following articles

-SutteARIMA: Short-term forecasting method, a case: Covid-19 and stock market in Spain

-Forecasting the prevalence of COVID-19 outbreak in Egypt using nonlinear autoregressive artificial neural networks

-Application of the ARIMA model on the COVID-2019 epidemic dataset

Reviewer #3: 1- What is the difference between ARIMA and ARMA?

2- What are the benefits of your model over "nonlinear autoregressive artificial neural networks" used in

Forecasting the prevalence of COVID-19 outbreak in Egypt using nonlinear autoregressive artificial neural networks

3- What is RMSE, MAE, COV, CRM of the obtained results?

4- Historical total and daily confirmed cases should be included.

5-Add world cases in Fig 8

6- Plot your data as a time series in stead of using single day data

6. PLOS authors have the option to publish the peer review history of their article (what does this mean?). If published, this will include your full peer review and any attached files.

Reviewer #1: No

Reviewer #2: No

Reviewer #3: No

---

## [Author Response · Author response to Decision Letter 0]

26 Jun 2020

Please see attached a file named response to reviewer. All modifications and responses are explained in a word file attached with this R1 submission.

---

## [Decision Letter · Decision Letter 1]

6 Jul 2020

Assessment of the outbreak risk, mapping and infection behavior of COVID-19: Application of the autoregressive integrated-moving average (ARIMA) and polynomial models

PONE-D-20-10865R1

Dear Dr. Heidari,

We’re pleased to inform you that your manuscript has been judged scientifically suitable for publication and will be formally accepted for publication once it meets all outstanding technical requirements.

Kind regards,

Jie Zhang

Academic Editor

PLOS ONE

Additional Editor Comments (optional):

Reviewers' comments:

Reviewer's Responses to Questions

**Comments to the Author**

1. If the authors have adequately addressed your comments raised in a previous round of review and you feel that this manuscript is now acceptable for publication, you may indicate that here to bypass the “Comments to the Author” section, enter your conflict of interest statement in the “Confidential to Editor” section, and submit your "Accept" recommendation.

Reviewer #2: All comments have been addressed

Reviewer #3: All comments have been addressed

2. Is the manuscript technically sound, and do the data support the conclusions?

Reviewer #2: Yes

Reviewer #3: Yes

3. Has the statistical analysis been performed appropriately and rigorously? 

Reviewer #2: Yes

Reviewer #3: Yes

4. Have the authors made all data underlying the findings in their manuscript fully available?

Reviewer #2: Yes

Reviewer #3: Yes

5. Is the manuscript presented in an intelligible fashion and written in standard English?

Reviewer #2: Yes

Reviewer #3: Yes

6. Review Comments to the Author

Reviewer #2: All comments has been taken into consideration. The revised manuscript is ready for publication in PLOS ONE.

Reviewer #3: "Assessment of the outbreak risk, mapping and infection behavior of COVID-19:

Application of the autoregressive integrated-moving average (ARIMA) and polynomial

models" has been revised in a good manner.

7. PLOS authors have the option to publish the peer review history of their article (what does this mean?). If published, this will include your full peer review and any attached files.

Reviewer #2: No

Reviewer #3: No

---

## [Editor Report · Acceptance letter]

10 Jul 2020

PONE-D-20-10865R1 

Assessment of the outbreak risk, mapping and infection behavior of COVID-19: Application of the autoregressive integrated-moving average (ARIMA) and polynomial models 

Dear Dr. Heidari:

I'm pleased to inform you that your manuscript has been deemed suitable for publication in PLOS ONE. Congratulations! Your manuscript is now with our production department. 

Kind regards, 

on behalf of

Dr. Jie Zhang 

Academic Editor

PLOS ONE